# CdS Nanoparticles Supported by Cobalt@Carbon-Derived MOFs for the Improved Adsorption and Photodegradation of Ciprofloxacin

**DOI:** 10.3390/ijms241411383

**Published:** 2023-07-13

**Authors:** Mi Li, Qin Fang, Yan Lai, Luying Chen, Qiucheng Fu, Jiao He, Yongjuan Chen, Liang Jiang, Zhiying Yan, Jiaqiang Wang

**Affiliations:** 1School of Chemical Sciences & Technology, Yunnan University, Kunming 650091, China; lm8827@outlook.com (M.L.); fangqin1017@163.com (Q.F.); laiyan0129@163.com (Y.L.); cly159365128122022@163.com (L.C.); 2548758390@mail.ynu.edu.cn (Q.F.); hejiao@ynu.edu.cn (J.H.); chenyongjuan@ynu.edu.cn (Y.C.); jiangliang@ynu.edu.cn (L.J.); 2Yunnan Province Engineering Research Center of Photocatalytic Treatment of Industrial Wastewater, Yunnan University, Kunming 650091, China; 3School of Materials & Energy, Yunnan University, Kunming 650091, China

**Keywords:** CdS/Co@C, ciprofloxacin oxidation, metal–organic frameworks, visible-light photocatalysis

## Abstract

The design and synthesis of efficient photocatalysts that promote the degradation of organic pollutants in water have attracted extensive attention in recent years. In this work, CdS nanoparticles are grown in situ on Co@C derived from metal–organic frameworks. The resulting hierarchical CdS/Co@C nanostructures are evaluated in terms of their adsorption and photocatalytic ciprofloxacin degradation efficiency under visible-light irradiation. The results show that, apart from offering a large surface area (55.69 m^2^·g^−1^), the prepared material can effectively suppress the self-agglomeration of CdS and enhance the absorption of visible light. The CdS/Co@C-7 composite containing 7% wt Co@C has the highest photodegradation rate, and its activity is approximately 4.4 times greater than that of CdS alone. Moreover, this composite exhibits remarkable stability after three successive cycles of photocatalysis. The enhanced photocatalytic performance is largely ascribed to the rapid separation of electron–hole pairs and the effective electron transfer between CdS and Co@C, which is confirmed via electrochemical experiments and photoluminescence spectra. The active substance capture experiment and the electron spin resonance technique show that h^+^ is the main active entity implicated in the degradation of CIP, and accordingly, a possible mechanism of CIP photocatalytic degradation over CdS/Co@C is proposed. In general, this work presents a new perspective on designing novel photocatalysts that promote the degradation of organic pollutants in water.

## 1. Introduction

Ciprofloxacin (CIP) is a broad-spectrum antibiotic that is widely used in medicine, aquaculture, and agricultural production. However, the incomplete metabolization of this antibiotic in humans and animals leads to high CIP concentrations in surface water, which inevitably affects aquatic ecosystems and human health. Photocatalytic degradation is considered a green technology that can effectively remove pollutants from water [1]. As a typical and important semiconductor, cadmium sulfide (CdS) is highly attractive in terms of the photocatalytic purification of environmental contaminants owing to its narrow band gap (2.4 eV) and excellent visible-light response. However, CdS alone are prone to the insufficient separation of photogenerated charge carriers, particle aggregation, and photocorrosion, which seriously restrict their activity as well as practical applications on a large scale [2]. In order to overcome the inherent defects of a single material, various efforts have been made to modify the structure and composition of CdS [3,4,5]. Among these strategies, coupling CdS with carbon-based materials has been proven to be effective and economical in improving photocatalytic efficiency. The carbon-based material, especially composites of carbon-coated low-cost metals (M@C), have attracted widespread attention due to their high conductivity and unique structure [6,7,8,9,10]. The composites bear the properties of the metal and carbon as cocatalysts and significantly improve the catalytic activity of CdS [6,7,8]. Moreover, the encapsulation of transition metal particles in the surface carbon layer can effectively improve the oxidation resistance of metals and makes the structure more stable. Moreover, recent studies have shown that metal–organic frameworks (MOFs) are effective sacrificial templates for preparing M@C hybrid materials via direct pyrolysis. MOF-derived M@C has the unique advantages of large surface area, highly electrical properties, high stability, and low metal ion leaching yield. For instance, Ren et al. [11] fabricated ultrathin graphene-coated Cu nanoparticles (Cu@C) via the two-step pyrolysis of MOF precursors, which solved the problem of poor oxidation resistance of Cu. And the obtained Cu@C/SrTiO_3_ composites achieve excellent sustained stability and photocatalytic activity. Fang et al. [12] prepared nickel nanoparticles wrapped up by graphene layers (Ni@C) via MOF pyrolysis and leaching of hydrochloric acid. As an efficient co-catalyst, it greatly improved the photocatalytic hydrogen evolution activity of g-C_3_N_4_. Notably, M@C derived from MOFs has shown the improved photocatalytic hydrogen evolution activity of CdS. However, this current research is confined to improving light capture, charge separation, and photostability of catalysts. In terms of photocatalytic remediation, due to the small specific surface area of CdS, the adsorption of organic pollutant molecules is also an important point affecting the photocatalytic activity of CdS. But, unfortunately, little attention has been paid to it.

Importantly, in addition to excellent chemical stability and high electrical properties, M@C inherits the characteristics of the original MOFs, including large surface areas, high porosities, and ordered pore structures, which can lead to their wide potential applications. The porosity can improve the utilization of solar lights because the lights entering the pores can be repeatedly reflected until they are possibly fully absorbed [13]. In addition, the large surface area ensures that M@C may adsorb more organic pollutant molecules, which helps improve the photocatalytic performance of CdS and the removal efficiency of organic pollutants. Yang et al. [14], via the pyrolysis of ZIF-8 and the growth of CdS, CdS/MOF-derived porous carbon (MPC) composites were prepared as highly efficient visible-driven photocatalysts, showing an excellent adsorption photocatalytic degradation of antibiotics. Hence, it is reasonable to believe that the assembly of CdS and M@C material may exhibit excellent adsorption and photocatalytic activity in water remediation. The large surface area of ZIF-8, as a precursor to prepare graphitic carbon, is conducive to the discrete growth of CdS and improves the situation that CdS is easy to agglomerate. However, the photocatalytic effect of CdS is limited. Considering that ZIF-67, which has a similar topological structure and cell parameters to ZIF-8 [15], is easily broken at high temperatures, core–shell ZIF-8@ZIF-67 is the precursor system to prepare graphite carbon nanocages coated with metal Co, which can introduce metal Co and a hollow cage structure, which is expected to further improve the photocatalytic activity of CdS. However, to the best of our knowledge, except for the photocatalytic hydrogen evolution, there is little work about the hybrids consisting of CdS and M@C for the removal of organic contaminants in water by far.

Herein, a hierarchical CdS/Co@C nanostructure is fabricated by growing CdS nanoparticles on Co@C polyhedrons derived from ZIF-8@ZIF-67. Its structure and properties are studied and discussed in detail. As expected, the obtained composite combines the advantages of Co@C and CdS and concurrently utilizes the synergistic effect between them, exhibiting excellent adsorption and photocatalytic activity towards CIP. Furthermore, efforts have been made to elucidate the reasons and mechanisms for the enhanced photocatalytic activity of CdS/Co@C composites.

## 2. Results and Discussion

### 2.1. Characterization

The crystal structure and phase of the synthesized materials were investigated using XRD. Compared to the single ZIF-8 compound, the ZIF-8@ZIF-67 composites exhibit similar diffraction peaks, and there are no additional peaks (Appendix A), which is consistent with the available literature [9]. As shown in Figure 1, Co@C presents a characteristic diffraction peak at 26.1° corresponding to the (002) plane of graphitic carbon. This reveals that Co@C is graphitized to a certain degree after calcination at 800 °C [16]. The Raman spectrum of Co@C (Appendix A) further confirms the graphitization of the material. The peaks observed at 44.2°, 51.5°, and 75.9° are attributed to the (111), (200), and (220) planes of metal Co (JCPDS no. 15-0806), respectively, which proves that the Co@C had been successfully synthesized [17]. Furthermore, the XRD patterns of all CdS/Co@C composites show characteristic diffraction peaks similar to those observed for CdS at 26.5°, 43.9°, and 52.0°, corresponding to the (111), (220), and (311) facets of cubic CdS (JCPDS no. 65-2887), respectively [18]. However, the peaks of CdS and Co@C in the CdS/Co@C XRD spectrum cannot be clearly distinguished, possibly due to the overlap of peaks. Considering that no new diffraction peaks were detected in the spectrum of CdS/Co@C, the synthesis process does not generate any by-products.

The morphologies and microstructures of the synthesized materials are shown in Appendix A. ZIF-8@ZIF-67 has a rhomboid dodecahedron structure with a smooth surface similar to ZIF-8 [15]. Notably, the Co@C composite obtained via carbonization and acid etching inherits the polyhedral shape of the parent ZIF-8@ZIF-67 crystals without collapse (Figure 2a,b and Appendix A). Nevertheless, the surface of Co@C is rough compared to the parent crystals, and a hollow thin shell structure is generated, probably due to the evaporation of Zn and the catalytic effect of Co nanoparticles at high temperatures [9]. As shown in Figure 2c, many Co nanoparticles are observed on the matrix shell of Co@C, and based on high-resolution transmission electron microscopy (HRTEM) analysis, the crystal plane spacings of 0.20 nm and 0.34 nm correspond to the Co (111) and the (002) crystal planes of graphitized carbon, respectively [17]. This confirms that the Co nanoparticles are encapsulated in several nitrogen-doped graphitic carbon layers, which prevents their aggregation and protects them from corrosion, leading to excellent electron transport capacity [19]. Figure 2d,e demonstrate that the formed CdS nanoparticles are uniformly dispersed on the surface of Co@C. This close contact between CdS and Co@C facilitates a rapid electron transfer through the interface, thereby enhancing the efficiency of separation between photogenerated electrons and holes. Consistently, the HRTEM images in Figure 2f present lattice spacings corresponding to the (111) and (200) planes of CdS and the (002) plane of graphitic carbon. The EDS spectra of CdS/Co@C-7 (Appendix A) reveal that the material comprises C, N, O, Co, Cd, and S elements, which indicates that the CdS/Co@C composite had been successfully synthesized.

To further analyze the microstructures of the photocatalysts, their N_2_ adsorption–desorption isotherms were plotted. As shown in Figure 3, Co@C exhibited strong adsorption capacity at lower relative pressure, indicating that micropores are present. The hysteresis loop observed at relative pressures between 0.4 and 0.9 suggests that the structure is mesoporous [20]. As for CdS, it has a nonporous or macroporous structure, as evidenced by the type IV isotherm, with no hysteresis loop obtained for the material. When Co@C is coupled with CdS, the shape and trend of the adsorption–desorption isotherm does not change significantly, irrespective of the mass of Co@C. However, the quantity adsorbed at P/P_0_ around 0.2 and 1.0 slightly increases with increasing amounts of Co@C. The Brunauer–Emmett–Teller (BET) surface area (S_BET_) and pore structure data of all samples are listed in Appendix A. Clearly, the S_BET_ of CdS/Co@C is larger than that of CdS, and it increases as the amount of Co@C increases due to the large specific surface area of these particles (288.34 m^2^·g^−1^). In general, the increased surface area may provide more catalytic sites, which is conducive to photocatalytic reactions [21].

X-ray photoelectron spectroscopy (XPS) analysis was carried out to investigate the composition and chemical state of the samples. The XPS spectra showing the elemental compositions of CdS, Co@C, and CdS/Co@C-7 are presented in Figure 4a. The C 1s spectrum of Co@C shown in Figure 4b can be divided into three sub-peaks at 284.7, 285.7, and 286.8 eV, which are ascribed to C-C, C-N, and C=O bonds, respectively [10]. The identification of C-N bonds at 285.7 eV confirms the successful doping of N elements in the carbon matrix. After coupling with CdS, the C 1s spectrum of CdS/Co@C-7 does not significantly change compared to that of Co@C. However, the three sub-peaks slightly shifted to more negative binding energies (284.6, 285.6, and 286.7 eV). As shown in Figure 4c, the N 1s spectrum of Co@C can also be divided into three sub-peaks with binding energies of 398.6, 399.5, and 400.9 eV, corresponding to the pyridine N, pyrrole N, and graphite N, respectively [22]. Notably, the N 1s peak is hardly observed in the XPS spectrum of CdS/Co@C-7 due to the low content of the N element and the overlap of signal peaks. The Co 2p spectrum of Co@C (Figure 4d) exhibits a pair of peaks attributed to metallic Co at 778.6 (Co 2p_3/2_) and 793.7 eV (Co 2p_1/2_), which agrees well with the TEM results. The doublet peaks observed at 781.5 and 795.6 eV are ascribed to Co 2p_3/2_ and Co 2p_1/2_ of the Co^2+^ ions, respectively, which indicates that Co species on the surface of the material may be oxidized by air. The peaks at 785.3 and 801.6 eV are satellite signals [23,24]. Interestingly, the changes in the binding energies of Co peaks in the CdS/Co@C-7 spectrum compared to the Co@C-7 spectrum are similar to the changes detected in the binding energies of C 1s peaks. The S 2p spectrum of CdS (Figure 4e) can be well fitted to two peaks at 161.6 and 162.8 eV, which are assigned to S 2p_3/2_ and S 2p_1/2_, respectively [3]. In the case of CdS/Co@C-7, these two peaks are shifted to 161.8 and 163.0 eV. Finally, the Cd 3d peaks of CdS located at 405.1 and 411.9 eV are attributed to Cd 3d_5/2_ and Cd 3d_1/2_, respectively (Figure 4f), and they are shifted to higher binding energies in the case of CdS/Co@C-7, similar to the S 2p peaks [25,26]. All the shifts in binding energy discussed above indicate that electrons are being transferred from CdS to Co@C at the interface of Co@C/CdS-7 [27].

Figure 5a compares the UV–Vis DRS spectra of the CdS, Co@C, and CdS/Co@C composites synthesized herein. The maximum absorption wavelength of CdS is about 540 nm, indicating that CdS is responsive to visible light. Compared to CdS, the absorption edges of the CdS/Co@C composite show slight red shifts toward longer wavelengths, and the absorption intensity at 540–800 nm is significantly stronger due to the wide spectral absorption of black Co@C. The band edges estimated using the Kubelka–Munk equation (Appendix A) and corresponding Tauc plots (Figure 5b) clearly show that the band gaps of CdS/Co@C composites (2.38–2.43 eV) are lower than that of CdS (2.44 eV) [28,29,30].

### 2.2. Photocatalytic Degradation of Ciprofloxacin

The photocatalytic activities of the prepared materials were evaluated based on their performance in the photocatalytic degradation of CIP solution under visible-light irradiation. The adsorption of all catalysts was assessed before illumination. As shown in Figure 6a,b, all investigated materials reach adsorption–desorption equilibrium within 60 min, and the adsorption of CIP on CdS/Co@C composites increases gradually with increasing Co@C content. This is attributed to the restraining effect of Co@C on the aggregation of CdS, which provides more exposed adsorption sites [31]. After 120 min of illumination, the concentration of CIP hardly changes in the absence of a photocatalyst, indicating that CIP cannot degrade spontaneously under visible light. Moreover, Co@C exhibits almost no photocatalytic activity toward CIP, and the activity of pure CdS is poor. Comparatively, the activities of the CdS/Co@C composites are much higher, and CdS/Co@C-7 presents the largest photocatalytic degradation yield of 68.06% and the highest CIP total removal yield of 94.37% after 120 min of irradiation. When the content of Co@C exceeds 7% wt, the excess Co@C will block incident visible light, thereby lowering the efficiency of CdS/Co@C for photocatalysis. Based on the corresponding kinetic curves shown in Figure 6c, the photocatalytic degradation of CIP may obey the following pseudo-first-order kinetic model: ln (C_0_/C_t_) = kt, where C_0_ and C_t_ are the initial concentration and the concentration at different illumination times, respectively, k is the kinetic rate constant, and t is the illumination time [32]. Compared to CdS and Co@C, the apparent rate constants (k) of all CdS/Co@C composites are high, as shown in Figure 6d. Indeed, the k value of CdS/Co@C-7 (0.02199 min^−1^) is 4.4 and 26.2 times higher than those of pure CdS (0.00495 min^−1^) and Co@C (0.00084 min^−1^), respectively.

According to previous reports, the initial pH value of the CIP solution significantly affects the adsorption and heterogeneous photocatalytic degradation of CIP since this antibiotic has four different pKa values (3.64, 5.05, 5.9, and 8.9) [33]. Herein, we show that the adsorption efficiency increases from 3.43 to 49.23% as the initial pH value increases between 3.0 and 9.0 (Appendix A), which indicates that adsorption is strongly dependent on the initial pH value. The photocatalytic activity is also closely related to initial pH, as evidenced by the apparent first-order rate constants (k) determined at different pHs shown in Figure 7b. At pH = 3.0, CIP is hardly degraded (k = 0.00015 min^−1^). Since all pKa values of CIP are higher than 3.0, the CIP molecule is protonated and positively charged [33,34]. With an isoelectric point at pH = 5.8, the surface of CdS/Co@C-7 is also positively charged (Appendix A). Therefore, the low adsorption and slow degradation of CIP at pH = 3.0 may be attributed to the electrostatic repulsion between CdS/Co@C-7 and the CIP molecule [35]. When the pH value is increased from 3.0 to 4.0, the photocatalytic activity increases sharply. Indeed, the k value determined at pH = 4.0 (0.01040 min^−1^) is 69.6 times higher than that determined at pH = 3.0. This may be related to the deprotonation of the N atom (pKa = 3.64) in the quinolone ring [33]. At pH 5.0~8.0, the photocatalytic activity of CdS/Co@C-7 is high and stable (k = 0.1789–0.02321 min^−1^) due to the existence of CIP as zwitterions [36]. Although electrostatic repulsion exists between negatively charged CdS/Co@C-7 and CIP at pH = 9.0, the photocatalytic activity of CdS/Co@C-7 remains high (k = 0.02024 min^−1^) at this pH, unlike the case of pH = 3.0. This is possibly related to the incomplete deprotonation of piperazine substituents (pKa = 8.89) [34,36].

The reusability and stability of a photocatalyst are important properties needed to expand its practical application. To evaluate these properties for CdS and CdS/Co@C-7, three cycle experiments were carried out. After each run, the catalyst was collected, washed several times with 1 mmol·L^−1^ HCl solution and anhydrous ethanol, then dried for the next run. The experimental results shown in Figure 8a demonstrate that the CIP degradation over CdS decreases significantly after three consecutive runs. However, the degradation over CdS/Co@C-7 remains high. At the same time, the XRD patterns (Figure 8b) and the XPS survey spectra (Figure 8c) of CdS/Co@C-7 exhibit no significant difference before and after the photocatalytic reaction. The slight shift toward the lower binding energy of Co 2p spectra in used CdS/Co@C-7, as provided in Figure 8d, may be ascribed to e-transfer [37]. These results indicate that the structure of the photocatalyst is maintained.

### 2.3. Possible Photocatalytic Mechanism

The photocatalytic activity of a catalyst mainly depends on the efficiency of electron-hole separation and electron migration. Herein, the recombination probability of photogenerated electron–hole pairs was analyzed using photoluminescence (PL). As shown in Figure 9a, the steady-state photoluminescence (PL) spectra display obvious PL quenching of CdS/Co@C-7 composite, indicating that this material can inhibit the recombination of photo-induced electron–hole pairs [38]. Meanwhile, the time-resolved PL (TRPL) spectra (Figure 9b) show that CdS/Co@C-7 composite exhibits a shorter average PL lifetime than that of CdS. The apparent PL quenching and lifetime diminution indicate the efficient charge movement from CdS to Co@C in CdS/Co@C-7 composite. The transfer efficiency of photo-induced electron–hole pairs was investigated using electrochemical impedance spectroscopy (EIS), and the resulting Nyquist plots show that under visible-light irradiation, the semicircle radius of CdS/Co@C-7 is significantly smaller than that of CdS (Figure 9c), which suggests that the former material can achieve more efficient photoelectric separation and migration [39]. The transient photocurrent response spectra shown in Figure 9d reveal that both, CdS and CdS/Co@C-7 can produce a stable reversible photocurrent under visible-light irradiation; however, the photocurrent density on the latter is significantly higher. This suggests that the coupling between Co@C and CdS can strengthen charge transfer [40]. Overall, these results indicate that the introduction of Co@C can remarkably improve the carrier separation efficiency of CdS.

The Mott–Schottky technique was employed to measure the flat-band potential (E_fb_) and determine the conductive type of the photocatalyst. As shown in Appendix A, the positive tangent slopes of the C^−2^/E curves indicate that CdS, CdS/Co@C-7, and Co@C are all n-type semiconductors with E_fb_ values of −1.18, −0.89, and −0.85 V vs. Ag/AgCl, respectively, which corresponds to −0.63, −0.34, and −0.30 V vs. NHE, respectively [41,42]. It has been previously reported that the E_fb_ of an n-type semiconductor is equal to its Fermi level (E_f_) [8]. Therefore, the E_f_ values of CdS, CdS/Co@C-7, and Co@C are −0.63, −0.34, and −0.30 V (vs. NHE), respectively. Moreover, since the E_f_ of the n-type semiconductor is more positive (by 0–0.2 V) than the conduction band edge potentials (E_CB_), the E_CB_ values of CdS and CdS/Co@C-7 are roughly estimated at −0.83 and −0.54 eV, respectively. Knowing that E_VB_ = E_CB_ + E_g_, the valence band edge potentials (E_VB_) of CdS and CdS/Co@C-7 are thus calculated to be 1.61 and 1.84 eV, respectively [43]. According to the energy band theory, the difference between the E_f_ values of CdS and Co@C in CdS/Co@C composites causes the electrons in CdS to flow into Co@C until an equilibrium in both E_f_ is reached [44]. Under visible-light irradiation, electrons are firstly photoexcited from the VB to the CB of CdS. Due to the adaptive energy level structures and positional relationship of the CdS and Co@C, the CB electrons of CdS transferred to Co@C, a material with metallic properties and more positive E_f_ [45]. As a result, a Schottky barrier forms at the CdS/Co@C interface, allowing only electrons to flow from CdS to Co@C while preventing electrons from recombining with holes in the reverse direction. Similar electron transfer mechanisms have been proposed for CdS/NC@Mo_2_N and WP-NC/g-C_3_N_4_ [46].

The effective separation of photogenerated carriers undoubtedly promotes the production of active species. The role of the main active species in the reaction system was evaluated by introducing scavengers such as isopropanol (IPA), p-benzoquinone (BQ), and ethylenediaminetetraacetic acid disodium salt (EDTA-2Na) into the reaction system in order to eliminate hydroxyl radicals (·OH), superoxide radicals (·O_2_^−^), and photogenerated holes (h^+^), respectively [47]. The results illustrated in Figure 10a show that in the presence of IPA, BQ, and EDTA-2Na, the removal yield of CIP over CdS/Co@C-7 decreases from 92.56 to 90.89%, 80.38%, and 26.26%, respectively. This indicates that h^+^ may be the main active species in the photocatalytic degradation of CIP. To further assess the reactive species produced during the photodegradation of CIP over CdS/Co@C-7, ·O_2_^−^, ·OH, and h^+^ were analyzed using the ESR technique. As shown in Figure 10b, the characteristic signal of DMPO-·OH adducts is not observed under light conditions nor under dark conditions, indicating that the h^+^ in CdS/Co@C-7 cannot oxidize H_2_O to ·OH since its potential (1.84 eV vs. NHE) is lower than that of H_2_O/·OH (2.27 eV vs. NHE) [48]. Meanwhile, the signals due to DMPO-·O_2_^−^ adducts appear under light conditions, as observed in Figure 10c, but not under dark conditions. This is mainly attributed to the fact that the E_CB_ of CdS/Co@C-7 (−0.48 eV vs. NHE) is more negative than the O_2_/·O_2_^−^ potential (−0.33 eV vs. NHE) [49]. Moreover, the intensity of TEMPO signals detected under visible-light irradiation is significantly lower than that of the signals detected in the dark (Figure 10d), probably due to the consumption of TEMPO via h^+^ from CdS [50].

Based on the obtained results, a possible mechanism of CIP photocatalytic degradation over CdS/Co@C composites was proposed. As shown in Figure 1, in the presence of visible-light irradiation, the photogenerated electrons are transferred from the VB to the CB of CdS; then, they move rapidly across the heterojunction interface and fasten to the surface of Co@C [11,44]. At the same time, the electrons in Co@C combine with the O_2_ adsorbed on their surface, resulting in ·O_2_^−^ active species that can rapidly decompose CIP [45]. Furthermore, although the h^+^ accumulated in the VB of CdS cannot oxidize H_2_O to ·OH, it can directly oxidize CIP to smaller molecules. Consequently, the CdS/Co@C composite synthesized in this study exhibits remarkable photocatalytic performance in terms of CIP degradation.

## 3. Materials and Methods

### 3.1. Reagents and Materials

All chemicals, purchased from Adamas Reagent Co., Ltd. (Shanghai, China), were used without purification.

### 3.2. Fabrication of Co@C and CdS/Co@C Composites

Co@C was prepared according to a previously reported method, with some modifications [9]. Briefly, the ZIF-8@ZIF-67 particles were firstly annealed at 500 °C for 2 h, then at 800 °C for another 2 h at a ramp rate of 5 °C min^−1^ in flowing N_2_ in a tube furnace. After natural cooling to room temperature, 100 mg of the resulting black powder was added to 500 mL of 4 mol·L^−1^ HCl solution, and the solution was strongly stirred at room temperature for 12 h. The precipitate was collected via centrifugation, washed with water several times, then vacuum dried at 60 °C for 24 h.

The CdS/Co@C photocatalyst was synthetized using a simple hydrothermal procedure. First, 10.08 mg Co@C, 120 mg CH_3_CSNH_2_, and 308.47 mg Cd (NO_3_)_2_·4H_2_O were dispersed in 75 mL of ethanol. After stirring for 10 min, the mixture was heated in a Teflon-lined autoclave at 180 °C for 3 h. The obtained composite was collected via centrifugation. Then, it was washed three times with ethanol and deionized water. Considering that the mass of Co@C in the composite accounts for 7% of the theoretical yield of CdS, the synthesized photocatalyst was labeled CdS/Co@C-7. Other photocatalysts with different masses of Co@C were synthesized via the same procedure (1%, 3%, 5%, 10%). For comparison, pure CdS with no Co@C was also prepared.

### 3.3. Photocatalytic Activity Measurement

The photocatalytic activity of the catalysts was evaluated in a quartz tube photoreactor (Nanjing Xujiang, Ltd., Nanjing, China). In a typical experiment, 10 mg of the photocatalyst was added to 50 mL CIP solution (20.0 mg·L^−1^). The pH value of the solution was adjusted by adding 0.10 mol·L^−1^ NaOH or 0.10 mol·L^−1^ HCl solution. Prior to illumination, the suspension was stirred for 60 min in the dark to reach the adsorption–desorption equilibrium state. Subsequently, it was irradiated with the light issued from a 350 W Xe lamp with a UV cutoff filter (λ ≥ 420 nm). Three-milliliter samples of the reaction suspension were collected at 20 min intervals during illumination. After filtration via a 0.22 μm membrane filter, the maximum absorption of the supernatant solution was measured at 276 nm using a Shimadzu UV-2600 spectrometer to determine the concentration of CIP. The CIP removal efficiency (%) was calculated according to Equation (1):Removal yield = (C_0_ − C_t_)/C_0_ × 100%, (1)
where C_0_ is the initial CIP concentration, and C_t_ is the residual concentration of CIP after a specific irradiation time (t).

## 4. Conclusions

In summary, the Co@C nanocages consisting of Co-embedded carbon nanocages were prepared via the pyrolysis of precursor ZIF-8@ZIF-67. Then, CdS/Co@C was successfully synthesized using a simple solvothermal method with Co@C as a precursor. As a support and cocatalyst of CdS, the Co@C nanocages improve the stability of Co atoms, suppress the self-agglomeration of CdS nanoparticles, offer abundant catalytic sites, and facilitate the separation and transfer of charges. Our results show that the CdS/Co@C material exhibits excellent performance in the photocatalytic degradation of ciprofloxacin. This is due to the high synergy between Co@C and CdS, which, in turn, is attributed to the unique structure and composition of Co@C. In addition, CdS/Co@C shows excellent stability better than pure CdS and can be reused, which has great prospects to apply in the fields of environmental and aquatic ecosystems. In general, this work constitutes a new reference for the design of high-performance photocatalysts that can effectively degrade pollutants.

## Data Availability

Data available on request.

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
