# Peer review of "CdS Nanoparticles Supported by Cobalt@Carbon-Derived MOFs for the Improved Adsorption and Photodegradation of Ciprofloxacin"

_ijms, 2023, doi:10.3390/ijms241411383_

Round 1
Reviewer 1 Report
J. Wang et al. presented CdS/Co@C nanostructures for ciprofloxacin photodestruction. The motivation for this study is quite clear from the discussion presented. However, the work is not presented in detail. There is no discussion and no section on the synthesis of CdS/Co@C nanostructures. Based on the data presented, it is difficult to confirm the presence of the reported composition. Since the data on the composition and structure of the composite are incomplete, there is a question about the correspondence of the title of the work to its content. The role of MOFs remains unclear. The photocatalytic part of the work also does not demonstrate novelty in this area, since there are reports in the literature on various catalytic systems that successfully perform similar applications.
Reviewer 2 Report
The paper entitled "CdS nanoclusters supported by Cobalt@Carbon derived MOFs for the improved adsorption and photodegradation of Ciprofloxacin" describes the preparation and utilization of new catalysts based on MOFs materials support which was tested in adsorption and photodegradation processes of ciprofloxacin.
Overall, this is a great work and the results are of high interest for this field. I identified only few minor issues to be addressed:
- in the title the authors mentioned about nanoclusters CdS. How they demonstrated?
-In Abstract, line 14, the surface are should be mentioned. At line 21-22-this perspective was before reported (see 10.1016/j.apsusc.2017.05.102), so that the novelty consists mainly in some innovations in the field.
-in Keywords: "Co nanoparticles Embedded Carbon" is mentioned only there, is not really a keyword, it should be reformulated.
-in Introduction, the authors should add the already published articles involving CdS-MOFs-based catalysts (see 10.1016/j.apsusc.2017.05.102) at lines 61-62. At line 64, the authors must motivate the choose of ZIF-8@ZIF-67 hybrid.
-in Results and Discussions in Fig. 1 the authors should explain the sample codes.
-at lines 142-143 how authors explain very small differences on the band gap and also the higher catalytic activity for the composite materials at these differences;
-the stability of ciprofloxacin at the studied pH values must be evaluated? If some aggregation of the drug occur?
-the residual water also must be analyzed by MS spectroscopy to identify the fragments and the mechanism elucidated based on these data;
- the Conclusions section is too general. There are important results that must be considered.
Based on these considerations, my recommendation is Minor revision.
Some grammar errors must be corrected.
Reviewer 3 Report
This article describes the immobilization of nanoclusters in Metal-Organic Frameworks towards showing an photocatalytic activity on a antibiotic. Nano-composite MOFs are an important research topic over 2 decades and still relevant due to its high catalytic ability, high porosity, ease of synthesis and heterogeneous nature. This article no different and successfully demonstrated all the traits of such composites. I would recommend its publication with some minor suggestions
1. Introduction needs to be more detail on describing the scope of work, and past literature. Since this field no new, and several other previous instances of such composites already exist in literature its important for authors to describe the novelty of this work and how it separates from the previous works done in ZIF based nanocomposites. Also I would suggest that the authors cite important papers from the group of Omar Yaghi, H.C Zhou, Jeff long, and S. Kitagawa to show the seminal past works done in the field
2. Authors have done PXRD, BET and SEM to show morphology and encapsulation of the nanoparticles in MOF pore, however is there any studies done to show if the nanoparticles are indeed encapsulated and not attaching to the surface of the MOF. May be a computation modeling would help to demonstrate the situation.
3. The authors did a nice job in XPS data analysis. I have a quick suggestion on UV experiment though Is there possible to show any visible color changes while encapsulation is happening. CdS may show a faint color but not sure?
4. Photocatalysis reactions are straightforward and done nicely. However, the assumption is that all surface of the MOF nanoparticle composite is the active site, is it true and do authors have any prove based on that? Hexagonal shape of MOF structure may alter the active sites and catalytic activity. There have been many previous examples of such study, a computation modeling might prove useful. If not maybe state, the assumption as well in the manuscript.
5. A chemical reaction will be useful for people from materials and engineering background
6. Nice work in mechanistic studies
Round 2
Reviewer 1 Report
The authors significantly improved the manuscript and addressed most of the reviewer's comments compared to its first version. Thus, I recommend the article to publish in its current form.